# In Situ Fatigue Damage Monitoring by Means of Nonlinear Ultrasonic Measurements

Andrea Saponaro and Riccardo Nobile *

Department of Engineering for Innovation, University of Salento, Via Per Monteroni, 73100 Lecce, Italy;
andrea.saponaro@unisalento.it
* Correspondence: riccardo.nobile@unisalento.it; Tel.: +39-0832-297771

**Abstract:** In the present work, the results of acoustic nonlinear response of ultrasonic wave propagation when monitoring the progress of damage induced by fatigue on notched C45 carbon steel specimens have been reported. Two ultrasound probes were fixed to the specimens during the tests. The input signal was sinusoidal type, while the corresponding ultrasound response signal was acquired and recorded at each stage of the test by means of a digital oscilloscope. A nonlinear frequency study was performed on the acquired data to evaluate the change in the second- and third-order nonlinearity coefficients of $\beta_1$ and $\beta_2$, respectively, on the tested specimens. Ultrasonic results were correlated to plastic strain at the notch tip in the initial phases of fatigue and stiffness degradation. The results showed a significant increase in second-order nonlinearity $\beta_1$ in the early stages of fatigue life. Subsequently, starting from about 30–40% of the fatigue life, the nonlinearity of $\beta_1$ increases. Before final failure, from 80 to 85% of fatigue life, the second-order nonlinearity further increases in the crack propagation stages. The nonlinear parameter of the third-order $\beta_2$ was less sensitive to damage than the parameter $\beta_1$, showing a rapid increase only starting from approximately 80 to 85% of the fatigue life. The proposed method proved to be valid for detective damage induced by fatigue and to predict the lifetime of metal materials.

**Keywords:** structural health monitoring (SHM); C45 steel; nonlinear ultrasonic; fatigue damage; harmonics





## 1. Introduction

Fatigue damage is one of the main failure mechanisms in metal structures. Mechanical components are often subject to variable loads over time, and the evaluation of their structural integrity for the entire life in service is an important challenge to overcome [1–4]. Fatigue strength is influenced by the manufacturing process of the material [5]. In this regard, the cold forming of metals can cause strong defects, such as distortion of the metal lattice with consequent shearing and sliding of the grains, as well as grain elongation. These defects affect conventional material properties, such as work hardening and residual stresses of the material. Both mechanical properties influence fatigue resistance [6]. Consequently, there is a growing demand for non-destructive testing, NDT, and structural monitoring techniques to monitor damage in service and prevent catastrophic failures by reducing maintenance costs [7–11]. The most ordinary checks performed in the industrial field to verify the integrity of the structures are based on ultrasounds, eddy currents, penetrating liquids, and magnetic particles. If the ultrasound measurement in linear condition constituted a standard and well-known experimental technique, the use of nonlinear ultrasound techniques is relatively uncommon, but it is significantly increasing due to its high sensitivity to micro-fatigue cracks compared to conventional methods [12–17].

The nonlinear ultrasound technique is based on the nonlinear elastic interaction between the material and the propagation of the ultrasound wave that passes through it. In the literature, this technique has been considered a potential non-destructive evaluation method for the study of the degradation phenomena in structures and components subject

to variable loads [18,19]. Moreover, studies on nonlinear techniques have shown that the first fatigue damage in metals is closely related to the nonlinear effects of ultrasound waves [20–23]. The ultrasound wave is substantially distorted during the propagation in the material such that it generates components of harmonics with a frequency greater than the fundamental, in which the amplitudes of the components of these harmonics depend on the elastic nonlinearity of the material. In other words, the method based on higher-order harmonic generation measures the amplitudes of the harmonic components after propagation in a material to evaluate the level of nonlinearity of that material. Second-order nonlinearity, which is the lowest of the higher-order nonlinearities, has been particularly considered because it is easier to measure than higher-order harmonics. Generally, nonlinear ultrasonic measurements use the generation of harmonics for the evaluation of the nonlinearity parameter of the material, most commonly using piezoelectric transducers for the excitation of ultrasonic waves thanks to their efficiency in generating high-amplitude wave packets [24–27]. However, as demonstrated in [28], the coupling of the transducers strongly affects the measurements of the amplitudes, and it is difficult to apply similar coupling conditions for different measurements. In this regard, several authors have proposed methods to reduce this variability [29,30]. To avoid conditions of variability in the coupling, it is preferable to have transducers permanently mounted on the structure for in situ fatigue monitoring. This also allows us to obtain a reliable measure based on the reference, thus avoiding the need for a calibration.

The second- and third-order nonlinear ultrasound parameters are related to the microstructure of the material and the internal micro-defects; therefore, they can be used to evaluate the microstructural changes induced by degradation, such as fatigue phenomenon. In a recent study [31], the authors showed a strong relationship between the variation of the acoustic response and the propagation of ultrasonic waves for the in situ monitoring of fatigue damage. From ultrasonic measurements carried out on a batch of specimens similar to those tested in this study, it has been shown that the attenuation of the received signal and of the fundamental frequency are more sensitive to damage than the time of flight and the degradation of the stiffness of the material. In other studies, it has been shown that microstructural changes due to the accumulation of damage introduce changes in the response of the material and can lead to variation in the nonlinearity parameters [32–36].

In a typical Structural Health Monitoring (SHM) setup, piezoelectric sensors are used to generate Lamb waves [24,26]. In this study, an original experimental setup was used to carry out ultrasonic nonlinear measurements on metallic specimens during fatigue tests. The novelty of the experimental setup is that conventional angled ultrasound probes, commonly used in traditional non-destructive testing, have been used. The configuration used allows us to obtain a wave that propagates in a predefined direction and, therefore, covers a well-localized area within the material, with the aim of monitoring any damage that occurs in the space between the two ultrasonic probes. This method is particularly interesting, especially when dealing with structures that have potential critical points at a local level such as notches, holes, etc., responsible, in most cases, for triggering the fatigue failure process. Another peculiarity of the proposed configuration is that by using transducers with rather high frequencies (2 MHz and 4 MHz), the recorded measurement is much less influenced by the noise due to external sources and to the test machine, producing a stable and accurate signal in every condition during the entire test.

The advantage of the proposed setup is that the ultrasonic measurement is much more accurate and precise due to less dispersion than the guided Lamb waves, whose speed depends on both the excitation frequency and the thickness of the plate where they propagate [37] but requires the use of multiple transducers to monitor various critical points that may be present in a structure. Guided Lamb waves are often used in non-destructive testing controls because they can travel long distances with little energy loss [38–42]. The main problem with such waves is that they are very dispersive (their phase and group velocities depend on the frequency) and multimodal (different movement of particles) [43]. Further-

more, the reflection mechanisms of Lamb waves are complex and difficult to understand, which limits the performance of the detection [44].

In the present paper, the nonlinear ultrasonic parameters were experimentally evaluated at various stages over the fatigue test on carbon steel C45 specimens subjected to different levels of load amplitude to monitor the evolution of fatigue damage. The fatigue tests are interrupted at different numbers of cycles with adequate sampling intervals, depending on the expected fatigue life, to perform the nonlinear ultrasonic measurements. The results of the nonlinear ultrasound parameters were correlated with the degradation of the stiffness and the plastic deformation of the material located at the tip of the notch by means of a simple FEM model, justifying the experimental evidence from a quantitative point of view. The applied experimental technique proved to be valid for studying the evolution of fatigue damage with non-destructive analysis and for predicting the fatigue life of metals effectively.

## 2. Review of Acoustic Nonlinearity Parameters

Ultrasonic nonlinear effects cause of the generation of higher harmonics in acoustic waves propagating through a material. Variations in the velocity of the ultrasound wave determine distortion of the traveling wave, which affects the structure of the wave frequency; for example, for an initially pure sine wave consisting of a single frequency, the peaks of the wave travel faster than valleys and the pulse becomes cumulatively more like a sawtooth waveform. In particular, the wave distorts by introducing other frequency components into the Fourier spectrum. Ultrasonic nonlinearity is characterized by the nonlinearity parameters $\beta_1$ and $\beta_2$, by which the amount is quantified that an ultrasonic wave is distorted as it travels through the specimen [45,46]. The wave is distorted due to the presence of defects and can be viewed as a measure of material quality.

The phenomenon of the detection mechanism of ultrasonic nonlinearity can be described by considering a wave of frequency f that travels through a metallic material. Experimentally, it has been observed that part of the acoustic energy of frequency f (fundamental component) transfers to its higher harmonics (2f, 3f and so on) that are generated during wave propagation. This effect is particularly useful as it allows for the detection of the early signs of damage since the transfer of energy to higher harmonic components occurs proportionally with the amount of damage.

Suppose that in a one-dimensional model, a single-frequency longitudinal wave propagates without undergoing attenuation. For small deformation, the oscillatory motion of the particles can be expressed by means of Equation (1) [45,46].

$$\rho \frac{\partial^2 u}{\partial t^2} = \frac{\partial \sigma}{\partial x} \tag{1}$$

where $\rho$ is the density of the material, $\sigma$ is the stress term and $u$ represent the displacement vector along the $x$ direction.

The constitutive equation of the nonlinear medium for Hooke's stress–strain behavior of a uniaxial stress state, can be described with Young's modulus $E$ by Equation (2):

$$\sigma = E f(\varepsilon) \tag{2}$$

By applying the power series development to Equation (2), the following equation is obtained [46]:

$$\sigma = E f(\varepsilon) = E\left(\varepsilon + \frac{1}{2}\beta_1\varepsilon^2 + \frac{1}{3}\beta_2\varepsilon^3 + \ldots + \frac{1}{n}\beta_{n-1}\varepsilon^n\right) \approx E\left(\varepsilon + \frac{1}{2}\beta_1\varepsilon^2 + \frac{1}{3}\beta_2\varepsilon^3\right) \tag{3}$$

where $n$ ($n = 1, 2, 3, \ldots$) is a factor representing the order of the nonlinear parameter of the material. Subsequently substituting Equation (3) in Equation (1) it can be grouped as Equation (4):

$$\rho \frac{\partial^2 u}{\partial t^2} = E \frac{\partial f(\varepsilon)}{\partial x} = E \frac{\partial}{\partial x} \left[ \varepsilon + \frac{1}{2} \beta_1 \varepsilon^2 + \frac{1}{3} \beta_2 \varepsilon^3 \right] \tag{4}$$

Positive strain is expressed by:

$$\varepsilon = \frac{\partial u}{\partial x} \tag{5}$$

Substituting Equation (5) into Equation (4), we have:

$$\rho \frac{\partial^2 u}{\partial t^2} = E \left[ \frac{\partial^2 u}{\partial x^2} + \beta_1 \left( \frac{\partial u}{\partial x} \right) \frac{\partial^2 u}{\partial x^2} + \beta_2 \left( \frac{\partial u}{\partial x} \right)^2 \frac{\partial^2 u}{\partial x^2} \right] \tag{6}$$

The relationship between the longitudinal wave velocity and Young's modulus is defined by [47]:

$$c = \sqrt{\frac{E}{\rho}} \tag{7}$$

therefore Equation (6) can be written as follows:

$$\frac{\partial^2 u}{\partial t^2} = c^2 \left[ \frac{\partial^2 u}{\partial x^2} + \beta_1 \left( \frac{\partial u}{\partial x} \right) \frac{\partial^2 u}{\partial x^2} + \beta_2 \left( \frac{\partial u}{\partial x} \right)^2 \frac{\partial^2 u}{\partial x^2} \right] \tag{8}$$

Assuming that the initial condition of Equation (6) [48,49] is given by:

$$u(0, t) = A_0 \sin \omega t \tag{9}$$

Applying the approximate perturbation method to solve Equation (8), in which only terms up to the second order have been considered [47], the solution can then be written as follows:

$$u(x, t) = A_0 \sin(\omega t - kx) + \frac{1}{8} \left( A_0^2 k^2 \beta_1 x \right) \cos 2(\omega t - kx) + \frac{1}{24} \left( A_0^3 k^3 \beta_2 x \right) [\cos 3(\omega t - kx) + 3 \cos(\omega t - kx)] \tag{10}$$

where $A_0$ is the fundamental harmonic ($A_0 = A_1$), K is the wave number and $x$ is the ultrasonic sound path. The second term has the frequency $2\omega$ and represents the second harmonic $A_2$ of the waveform that is generated by nonlinearities inhomogeneities present in the metallic medium. In the same way, the third term represents the third harmonic $A_3$ propagation in the material. The amplitudes of second and third terms can be written as follows [47,48]:

$$A_2 = \frac{1}{8} A_1^2 k^2 \beta_1 x \qquad A_3 = \frac{1}{24} A_1^3 k^3 \beta_2 x \tag{11}$$

Estimating the amplitudes of the first three harmonics $A_1$ (fundamental frequency), $A_2$ (second-order harmonic) and $A_3$ (third-order harmonic), it is possible to calculate the nonlinearity coefficients as follows:

$$\beta_1 = \left( \frac{A_2}{A_1^2} \right) \frac{1}{k^2 x} \qquad \beta_2 = \left( \frac{A_3}{A_1^3} \right) \frac{24}{k^3 x} \tag{12}$$

On the other hand, when $k$ and $x$ are constant, the absolute parameters $\beta_1$ and $\beta_2$ can be approximated as shown in Equation (13):

$$\beta_1 \propto \frac{A_2}{A_1^2} \qquad \beta_2 \propto \frac{A_3}{A_1^3} \tag{13}$$

The nonlinear parameters $\beta_1$ and $\beta_2$ contain essential information on the propagation of nonlinear wave.

## 3. Materials and Methods

The specimens used for fatigue testing were manufactured in normalized carbon steel (C45) with small notches in the centre of the gage length to localize the beginning of fatigue damage (Figure 1a). A preliminary static test, according to the ASTM E8-04 standard, was carried out on a smooth specimen on an MTS-810 (MTS Company, Swartz Creek, MI, USA) in displacement control with a velocity of 1 mm/min (Figure 1b) [31] and provided the tensile strength (UTS), the yield stress and Young's module (Table 1).

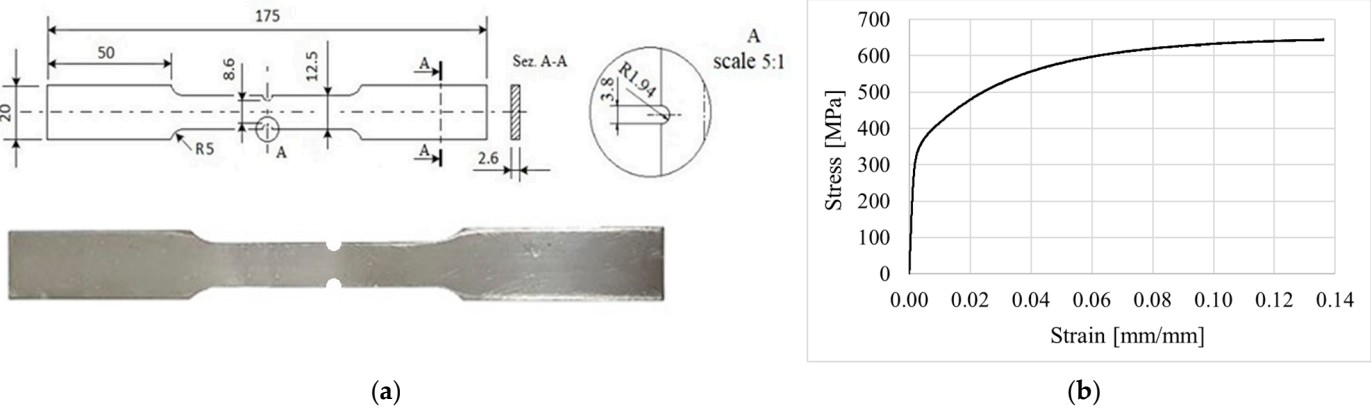

(a)                                                                                      (b)

**Figure 1.** (**a**) Notched specimen geometry (dimensions in mm) for nonlinear measurements; (**b**) stress/strain curve, adapted from [31].

**Table 1.** Experimentally determined mechanical properties.

| Mechanical Properties | |
| --- | --- |
| Ultimate Tensile Strength (UTS) | 640 [MPa] |
| 0.2% Yield Strength | 350 [MPa] |
| Young's Modulus | 219,000 [MPa] |

An in situ monitoring setup was used for nonlinear measurements while the specimen was undergoing fatigue cycles. Two commercial ultrasound probes (General Electric Company, Boston, MA, USA) were used in the transmission mode to monitor the progress of the damage induced by fatigue on notched steel specimens in situ. The ultrasonic probes, with the frequency of one double the other, respectively, 4 MHz and 2 MHz, were stably fixed to the specimens for the entire duration of the fatigue tests by means of steel brackets and bolts (Figure 2a,b) to avoid any possible variability due to the coupling with the specimen during the test. A stationary sinusoidal input signal was supplied to the transmitting probe by means of a function generator, while the received signal was acquired by the second probe at each step of the test and recorded using a digital oscilloscope. The recorded data were processed in a MATLAB software (R2016b version, MATLAB, produced by MathWorks, Inc. in USA) environment with an analysis algorithm based on the Fast Fourier Transform (FFT) to convert the nonlinear ultrasonic response in the time domain to the frequency domain. The attenuation of the ultrasonic signal was measured and compared with the beginning of the test signal, taken as a reference. Finally, a nonlinear frequency study was performed on the acquired data to evaluate the change in the second- and third-order nonlinearity parameters $\beta_1$ and $\beta_2$, respectively.

The experimental equipment used for nonlinear in situ ultrasonic measurements consists of a function generator (HAMEG Instruments GmbH, Mainhausen, Germany), a two-channel oscilloscope Agilent Keysight DSO-X-2012A (Keysight Technologies, Santa Rosa, CA, USA) with a maximum sample rate of 2GS/s, two commercial piezoelectric transducers (General Electric Company, Boston, MA, USA) with a 45° angled beam and a central frequency of 2 MHz (model WK 45 PB 2) and 4 MHz (model MWB 45-4) used

as a transmitter and receiver to excite and receive the longitudinal ultrasonic wave and a servo-hydraulic testing machine MTS-810 (MTS Company, Swartz Creek, MI, USA) with a load cell of 100 kN (Figure 2a). The receiver transducer had a central frequency that is two times that of the transmitter, which is needed to measure the second and third harmonic amplitude accurately.

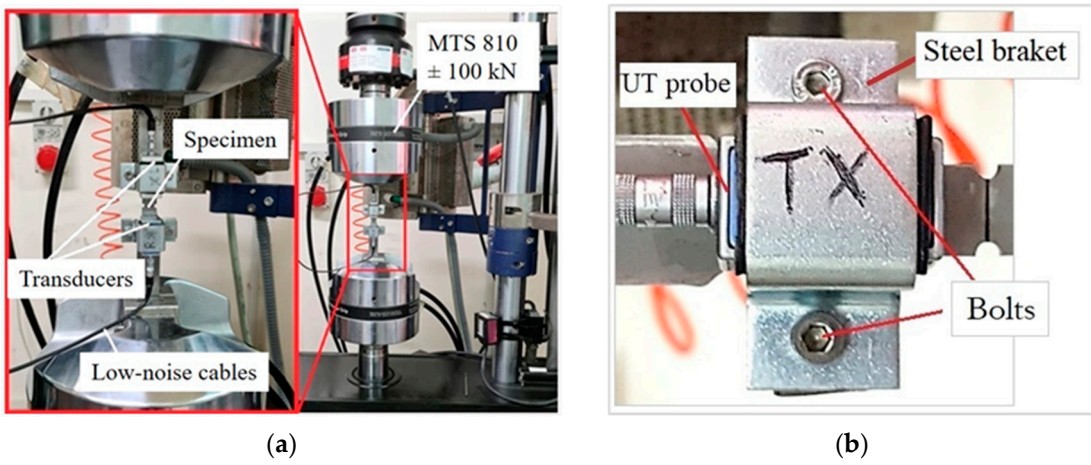

(**a**)                                                  (**b**)

**Figure 2.** Experimental setup of fatigue test: (**a**) specimen loaded on the MTS testing machine; (**b**) details on the steel brackets and bolts for stable fixing of the ultrasonic transducers on the specimen surface.

The pair of the two transducers were positioned symmetrically with respect to the notch section at a suitable distance between them of 41.6 mm, corresponding to an acoustic path of the longitudinal wave of 58.8 mm, for a total of sixteen reflections determined through the well-known Snell law (see Figure 3a), so that the angled ultrasound path beam covers the entire area where the notch is present [31]. The transducers are coupled to the specimen surface with lithium grease so that the coupling with the surface of the specimen remains stable during the entire duration of the test.

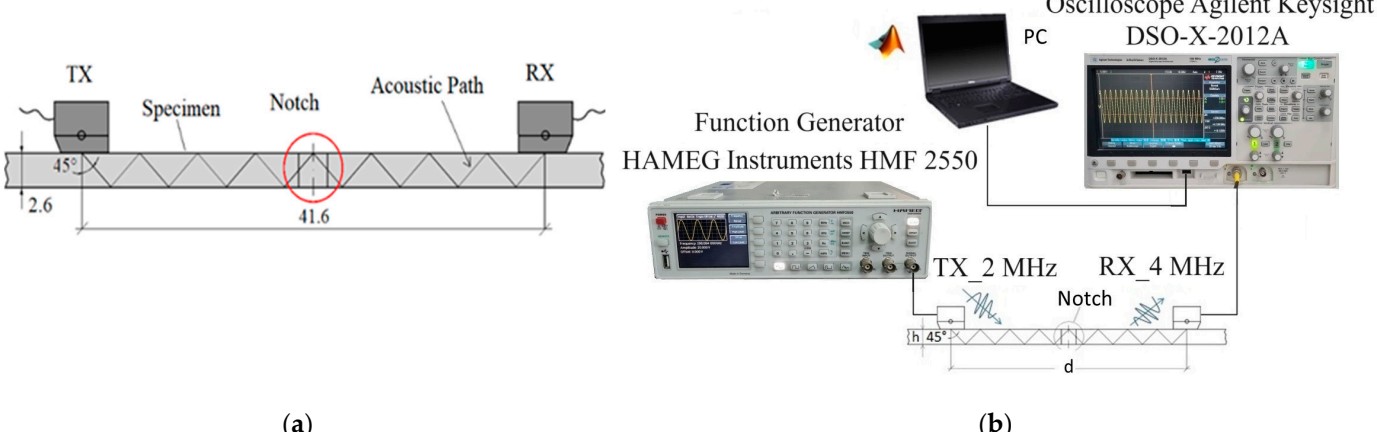

(**a**)                                                  (**b**)

**Figure 3.** (**a**) Position of transducers coupled to the specimen surface and acoustic patch (dimensions in mm); (**b**) schematic layout and devices of the nonlinear ultrasonic measurements (d = 41.6 mm and h = 2.6 mm).

A continuous sine waveform with a frequency of 2 MHz and 20 $V_{pp}$ (volt peak-to-peak) was produced by using a function generator (Hameg Instruments HMF2550, manufactured by HAMEG Instruments GmbH, Mainhausen, Germany) and sent to the transmitter while the receiving transducers were connected to the digital oscilloscope. A schematic block

diagram with the instruments used to measure the nonlinear ultrasound response in the experiments in transmission mode is shown in Figure 3b.

The received signals are recorded with a sampling rate of 500 GS/s, averaging 1024 times, and then transferred to a computer for further signal processing into the frequency domain using a MATLAB code based on Fast Fourier Transform (FFT). In this way, the amplitudes of the fundamental harmonic of the 2nd and 3rd order harmonic were determined, and the nonlinear coefficients were calculated according to Equation (13).

The specimens were subjected to tension–tension loading with a sinusoidal waveform, stress ratio R = $\sigma_{min}/\sigma_{max}$ = 0.1 and load frequency of 10 cycles per second. Four different load conditions were considered. Table 2 shows each fatigue test to the applied load in terms of maximum stress ($\sigma_{max}$) and stress amplitude ($\sigma_a$) and the number of cycles to failure ($N_f$).

**Table 2.** Scheduled experimental parameters for each fatigue test.

| Specimen ID | $\sigma_{max}$ [MPa] | $\sigma_a$ [MPa] | $N_f$ (Cycles) |
|:---:|:---:|:---:|:---:|
| $P_1$ | 335.42 | 150.93 | 183,922 |
| $P_2$ | 348.85 | 156.98 | 104,206 |
| $P_3$ | 357.78 | 161 | 97,297 |
| $P_4$ | 375.67 | 169.05 | 77,581 |

## 4. Experimental Results and Discussion

A direct comparison of the ultrasound signal recorded before the test and at a fixed number of cycles allows us to establish the existence of material behavior change originating from fatigue damage. For example, the ultrasound signals at 0 and 88,500 cycles of the $P_3$ specimen, corresponding to about 91% of fatigue life, are reported in Figure 4, both in the time and frequency domain. In the time domain, the peak–peak ultrasonic signals ($\Delta V_{pp}$) reduced from 17.66 mV to 16.99 mV (Figure 4a), while the fundamental harmonic $A_1$, accompanied by the second and third harmonic $A_2$, $A_3$, respectively, is clearly traced in the frequency domain (Figure 4b). Moreover, in the frequency domain, the second harmonic has an opposite behavior with respect to $A_1$ against fatigue damage, as $A_2$ is increased in comparison to the starting value.

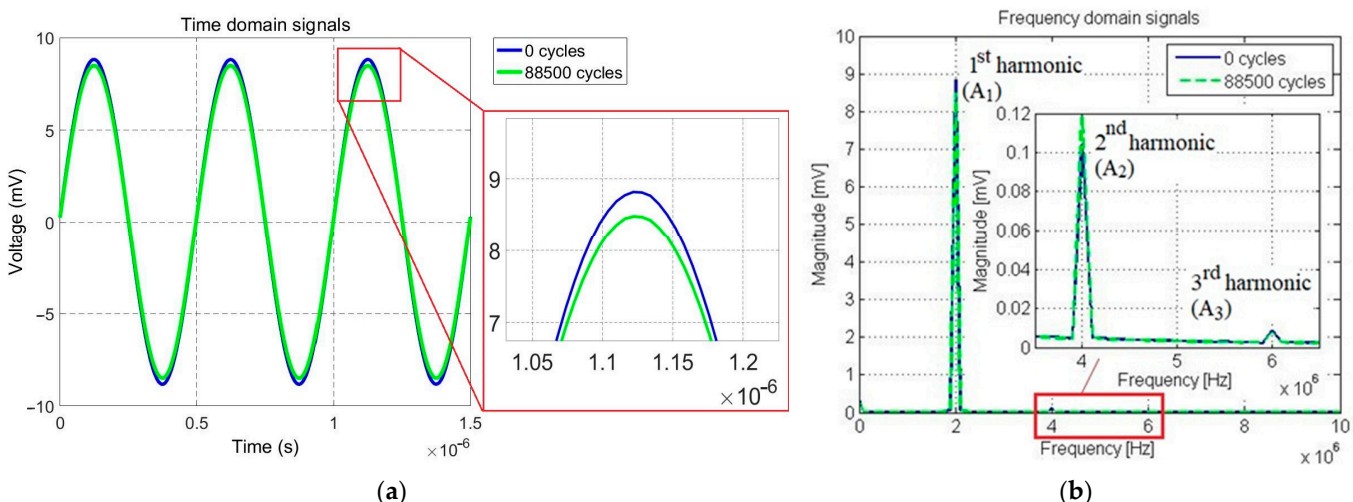

**Figure 4.** (**a**) Example of sinusoidal ultrasonic signals received in the time domain; (**b**) Fourier spectra (FFT) at 0 cycles and 88,500 cycles (91% of fatigue life) for specimen $P_3$.

The change in ultrasound signal is also evident in the nucleation phase. Figure 5 reports the ultrasound signals in the frequency domain corresponding to 0%, 31%, 62%,

and 91% of fatigue life, and it shows that the first harmonic initially decreases up to 31% and then increases up to failure. On the other hand, the second harmonic increases, starting from the beginning. Finally, at a fatigue life higher than 91%, in which a fatigue crack is already nucleated and is propagating, the change in the two harmonics is swinging and difficult to interpret.

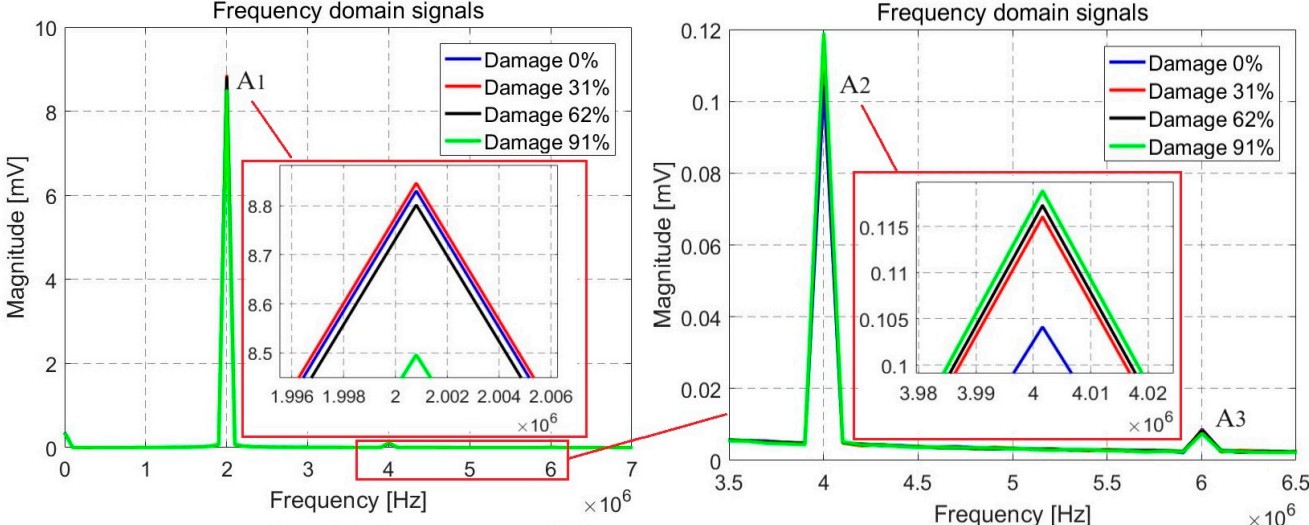

**Figure 5.** Variation of the first and the second harmonic in the FFT spectrum at different percentages of damage: 31%, 62% and 91% for the specimen $P_3$.

The behavior that has been described before for the $P_3$ specimen is common to the other specimens that have been tested at different stress amplitudes. The complete description of the experimental results is presented in Tables 3–6. Each table refers to a single specimen and reports the values of the three harmonics $A_1$, $A_2$, $A_3$ and $\beta_1$, $\beta_2$ parameters for each load cycle interval.

**Table 3.** Experimental nonlinear ultrasound parameters for specimen $P_1$.

| Number of Cycles ($N$) | Fatigue Life Ratio $N_i/N_f$ (%) | $A_1$ [mV] | $A_2$ [mV] | $A_3$ [mV] | $\beta_1$ [mV$^{-1}$] | $\beta_2$ [mV$^{-2}$] |
|---|---|---|---|---|---|---|
| 0 | 0 | 8.8326 | 0.114 | $8.48 \times 10^{-3}$ | $1.460 \times 10^{-3}$ | $1.231 \times 10^{-5}$ |
| 38,000 | 20.7 | 8.847 | 0.116 | $8.38 \times 10^{-3}$ | $1.482 \times 10^{-3}$ | $1.211 \times 10^{-5}$ |
| 57,000 | 31 | 8.834 | 0.1159 | $8.52 \times 10^{-3}$ | $1.485 \times 10^{-3}$ | $1.236 \times 10^{-5}$ |
| 76,000 | 41.3 | 8.81 | 0.1168 | $8.21 \times 10^{-3}$ | $1.505 \times 10^{-3}$ | $1.201 \times 10^{-5}$ |
| 94,000 | 51.1 | 8.806 | 0.1168 | $7.88 \times 10^{-3}$ | $1.506 \times 10^{-3}$ | $1.153 \times 10^{-5}$ |
| 104,000 | 56.5 | 8.802 | 0.1173 | $8.20 \times 10^{-3}$ | $1.514 \times 10^{-3}$ | $1.202 \times 10^{-5}$ |
| 132,500 | 72 | 8.801 | 0.1176 | $8.15 \times 10^{-3}$ | $1.518 \times 10^{-3}$ | $1.195 \times 10^{-5}$ |
| 151,000 | 82.1 | 8.81 | 0.1171 | $8.44 \times 10^{-3}$ | $1.509 \times 10^{-3}$ | $1.235 \times 10^{-5}$ |
| 160,600 | 87.3 | 8.821 | 0.1171 | $7.89 \times 10^{-3}$ | $1.505 \times 10^{-3}$ | $1.150 \times 10^{-5}$ |
| 167,000 | 90.8 | 8.813 | 0.1184 | $8.38 \times 10^{-3}$ | $1.524 \times 10^{-3}$ | $1.224 \times 10^{-5}$ |
| 175,000 | 95.1 | 8.7454 | 0.1184 | $8.09 \times 10^{-3}$ | $1.549 \times 10^{-3}$ | $1.210 \times 10^{-5}$ |
| 179,500 | 97.6 | 8.5036 | 0.1185 | $7.68 \times 10^{-3}$ | $1.639 \times 10^{-3}$ | $1.249 \times 10^{-5}$ |
| 181,500 | 98.7 | 8.43074 | 0.1164 | $7.79 \times 10^{-3}$ | $1.638 \times 10^{-3}$ | $1.300 \times 10^{-5}$ |
| 182,000 | 98.9 | 8.0405 | 0.1114 | $7.10 \times 10^{-3}$ | $1.723 \times 10^{-3}$ | $1.366 \times 10^{-5}$ |
| 183,500 | 99.8 | 7.1407 | 0.1024 | $5.21 \times 10^{-3}$ | $2.008 \times 10^{-3}$ | $1.431 \times 10^{-5}$ |

**Table 4.** Experimental nonlinear ultrasound parameters for specimen $P_2$.

| Number of Cycles (*N*) | Fatigue Life Ratio $N_i/N_f$ (%) | $A_1$ [mV] | $A_2$ [mV] | $A_3$ [mV] | $\beta_1$ [mV$^{-1}$] | $\beta_2$ [mV$^{-2}$] |
|---|---|---|---|---|---|---|
| 0 | 0 | 8.6261 | 0.1012 | $9.16 \times 10^{-3}$ | $1.360 \times 10^{-3}$ | $1.427 \times 10^{-5}$ |
| 5000 | 4.8 | 8.5996 | 0.1075 | $8.70 \times 10^{-3}$ | $1.454 \times 10^{-3}$ | $1.368 \times 10^{-5}$ |
| 10,000 | 9.6 | 8.6335 | 0.1088 | $8.73 \times 10^{-3}$ | $1.460 \times 10^{-3}$ | $1.356 \times 10^{-5}$ |
| 20,000 | 19.2 | 8.6934 | 0.1103 | $9.29 \times 10^{-3}$ | $1.460 \times 10^{-3}$ | $1.413 \times 10^{-5}$ |
| 30,000 | 28.8 | 8.7067 | 0.1109 | $9.57 \times 10^{-3}$ | $1.463 \times 10^{-3}$ | $1.450 \times 10^{-5}$ |
| 40,000 | 38.4 | 8.7270 | 0.1125 | $9.21 \times 10^{-3}$ | $1.477 \times 10^{-3}$ | $1.386 \times 10^{-5}$ |
| 50,000 | 48 | 8.7365 | 0.1130 | $9.37 \times 10^{-3}$ | $1.481 \times 10^{-3}$ | $1.406 \times 10^{-5}$ |
| 60,000 | 57.6 | 8.7416 | 0.1132 | $9.36 \times 10^{-3}$ | $1.481 \times 10^{-3}$ | $1.401 \times 10^{-5}$ |
| 70,000 | 67.2 | 8.7616 | 0.1135 | $9.29 \times 10^{-3}$ | $1.479 \times 10^{-3}$ | $1.381 \times 10^{-5}$ |
| 80,000 | 76.8 | 8.7645 | 0.1152 | $9.38 \times 10^{-3}$ | $1.499 \times 10^{-3}$ | $1.393 \times 10^{-5}$ |
| 90,000 | 86.4 | 8.6623 | 0.1134 | $9.04 \times 10^{-3}$ | $1.511 \times 10^{-3}$ | $1.391 \times 10^{-5}$ |
| 92,000 | 88.3 | 8.4349 | 0.1130 | $8.19 \times 10^{-3}$ | $1.588 \times 10^{-3}$ | $1.364 \times 10^{-5}$ |
| 94,000 | 90.2 | 7.9600 | 0.1140 | $6.54 \times 10^{-3}$ | $1.799 \times 10^{-3}$ | $1.297 \times 10^{-5}$ |
| 102,000 | 98 | 7.0507 | 0.0983 | $6.77 \times 10^{-3}$ | $1.978 \times 10^{-3}$ | $1.931 \times 10^{-5}$ |

**Table 5.** Experimental nonlinear ultrasound parameters for specimen $P_3$.

| Number of Cycles (*N*) | Fatigue Life Ratio $N_i/N_f$ (%) | $A_1$ [mV] | $A_2$ [mV] | $A_3$ [mV] | $\beta_1$ [mV$^{-1}$] | $\beta_2$ [mV$^{-2}$] |
|---|---|---|---|---|---|---|
| 0 | 0 | 8.831 | 0.1041 | $8.66 \times 10^{-3}$ | $1.335 \times 10^{-3}$ | $1.257 \times 10^{-5}$ |
| 10,000 | 10.3 | 8.827 | 0.1143 | $8.38 \times 10^{-3}$ | $1.467 \times 10^{-3}$ | $1.218 \times 10^{-5}$ |
| 20,000 | 20.5 | 8.842 | 0.1156 | $8.20 \times 10^{-3}$ | $1.479 \times 10^{-3}$ | $1.186 \times 10^{-5}$ |
| 25,000 | 25.7 | 8.841 | 0.1159 | $8.48 \times 10^{-3}$ | $1.483 \times 10^{-3}$ | $1.227 \times 10^{-5}$ |
| 30,000 | 30.8 | 8.847 | 0.116 | $8.38 \times 10^{-3}$ | $1.482 \times 10^{-3}$ | $1.211 \times 10^{-5}$ |
| 40,000 | 41.1 | 8.834 | 0.1159 | $8.52 \times 10^{-3}$ | $1.485 \times 10^{-3}$ | $1.236 \times 10^{-5}$ |
| 50,000 | 51.4 | 8.81 | 0.1168 | $8.21 \times 10^{-3}$ | $1.505 \times 10^{-3}$ | $1.201 \times 10^{-5}$ |
| 55,000 | 56.5 | 8.806 | 0.1168 | $7.88 \times 10^{-3}$ | $1.506 \times 10^{-3}$ | $1.153 \times 10^{-5}$ |
| 60,000 | 61.7 | 8.802 | 0.1173 | $8.20 \times 10^{-3}$ | $1.514 \times 10^{-3}$ | $1.202 \times 10^{-5}$ |
| 65,000 | 66.8 | 8.801 | 0.1176 | $8.15 \times 10^{-3}$ | $1.518 \times 10^{-3}$ | $1.195 \times 10^{-5}$ |
| 70,156 | 72.1 | 8.81 | 0.1171 | $8.44 \times 10^{-3}$ | $1.509 \times 10^{-3}$ | $1.235 \times 10^{-5}$ |
| 75,000 | 77.1 | 8.821 | 0.1171 | $7.89 \times 10^{-3}$ | $1.505 \times 10^{-3}$ | $1.150 \times 10^{-5}$ |
| 80,000 | 82.2 | 8.813 | 0.1184 | $8.38 \times 10^{-3}$ | $1.524 \times 10^{-3}$ | $1.224 \times 10^{-5}$ |
| 85,000 | 87.4 | 8.726 | 0.1186 | $7.98 \times 10^{-3}$ | $1.557 \times 10^{-3}$ | $1.201 \times 10^{-5}$ |
| 88,500 | 90.9 | 8.496 | 0.1188 | $7.55 \times 10^{-3}$ | $1.646 \times 10^{-3}$ | $1.231 \times 10^{-5}$ |
| 92,511 | 95.1 | 8.414 | 0.1168 | $7.74 \times 10^{-3}$ | $1.650 \times 10^{-3}$ | $1.299 \times 10^{-5}$ |
| 95,000 | 97.6 | 8.019 | 0.1117 | $6.98 \times 10^{-3}$ | $1.737 \times 10^{-3}$ | $1.353 \times 10^{-5}$ |
| 96,000 | 98.7 | 7.801 | 0.1034 | $6.37 \times 10^{-3}$ | $1.699 \times 10^{-3}$ | $1.342 \times 10^{-5}$ |
| 96,331 | 99 | 7.141 | 0.1024 | $5.21 \times 10^{-3}$ | $2.008 \times 10^{-3}$ | $1.431 \times 10^{-5}$ |

**Table 6.** Experimental nonlinear ultrasound parameters for specimen $P_4$.

| Number of Cycles ($N$) | Fatigue Life Ratio $N_i/N_f$ (%) | $A_1$ [mV] | $A_2$ [mV] | $A_3$ [mV] | $\beta_1$ [mV$^{-1}$] | $\beta_2$ [mV$^{-2}$] |
|---|---|---|---|---|---|---|
| 0 | 0 | 14.04 | 0.2879 | $6.53 \times 10^{-3}$ | $1.460 \times 10^{-3}$ | $2.360 \times 10^{-6}$ |
| 10,000 | 12.9 | 13.86 | 0.3101 | $6.60 \times 10^{-3}$ | $1.614 \times 10^{-3}$ | $2.480 \times 10^{-6}$ |
| 20,000 | 25.8 | 13.86 | 0.3096 | $6.40 \times 10^{-3}$ | $1.612 \times 10^{-3}$ | $2.402 \times 10^{-6}$ |
| 30,000 | 38.7 | 13.85 | 0.31 | $6.00 \times 10^{-3}$ | $1.616 \times 10^{-3}$ | $2.259 \times 10^{-6}$ |
| 40,000 | 51.5 | 13.84 | 0.3113 | $6.24 \times 10^{-3}$ | $1.625 \times 10^{-3}$ | $2.355 \times 10^{-6}$ |
| 45,506 | 58.6 | 13.89 | 0.3107 | $6.03 \times 10^{-3}$ | $1.610 \times 10^{-3}$ | $2.249 \times 10^{-6}$ |
| 50,000 | 64.4 | 13.91 | 0.3119 | $5.52 \times 10^{-3}$ | $1.612 \times 10^{-3}$ | $2.052 \times 10^{-6}$ |
| 60,000 | 77.3 | 13.8 | 0.3132 | $6.67 \times 10^{-3}$ | $1.645 \times 10^{-3}$ | $2.537 \times 10^{-6}$ |
| 65,000 | 83.8 | 13.2 | 0.3115 | $5.26 \times 10^{-3}$ | $1.788 \times 10^{-3}$ | $2.287 \times 10^{-6}$ |
| 68,000 | 87.6 | 12.8 | 0.3141 | $5.21 \times 10^{-3}$ | $1.917 \times 10^{-3}$ | $2.482 \times 10^{-6}$ |
| 70,000 | 90.2 | 12.52 | 0.3118 | $5.95 \times 10^{-3}$ | $1.989 \times 10^{-3}$ | $3.030 \times 10^{-6}$ |
| 72,000 | 92.8 | 12.44 | 0.3103 | $5.72 \times 10^{-3}$ | $2.005 \times 10^{-3}$ | $2.972 \times 10^{-6}$ |
| 74,000 | 95.4 | 12.18 | 0.3011 | $5.01 \times 10^{-3}$ | $2.03 \times 10^{-3}$ | $2.774 \times 10^{-6}$ |
| 77,000 | 99.2 | 9.342 | 0.2401 | $8.02 \times 10^{-3}$ | $2.751 \times 10^{-3}$ | $9.834 \times 10^{-6}$ |

The trend of the acoustic nonlinear parameters of the second and third order, normalized with respect to its initial value, is shown in Figure 6a,b. The first-order nonlinear parameter $\beta_1$ is interesting in terms of its continuous increase in relation to fatigue life (Figure 6a) in the nucleation phase, followed by a quick increase corresponding to the final phase immediately before the failure. This behavior allows us to identify crack nucleation with good accuracy. The behavior seems to be independent of the applied stress level. When observing the trend of the graphs relating to the tested specimens $P_2$, $P_3$ and $P_4$, an increase in the nonlinearity parameter in the initial loading phases is observed in all cases. This interesting behavior could be due to the initial plasticization in the tip of the notch due to the different load levels applied, distorting the ultrasonic wave and causing an evident increase in nonlinearity; in fact, from the trend of the curves, there is a greater increase in nonlinearity in the initial loading phases for specimen $P_4$ stressed with a greater load than specimens $P_2$ and $P_3$ stressed at a lower load level, crediting this hypothesis. This observation is coherent with some indications reported in the literature on stainless steels [49], in which the ultrasonic nonlinearity parameter of the second order increased with increasing plastic strain suffering under plastically deformed steel due to a microstructural change near the notch. The only difference was recorded in the initial trend of the $P_1$ specimen, which is not significant to the initial increase that characterizes the other specimens.

The third-order nonlinear parameter $\beta_2$, also considered in this work, did not show significant variation and was less sensitive to fatigue damage, in fact presenting stable values of up to about 95% of the fatigue life approximately equal to $1.2 \times 10^{-5}$ [mV$^{-2}$] for specimen $P_3$. Subsequently, it shows a significant increase in the crack propagation phase.

The trend of the third-order nonlinear parameter $\beta_2$, normalized with respect to its initial value assumed at 0 cycles, relative to all the tested samples (Figure 6b), shows a pattern of repeated behavior in the data, which is approximately constant up to about 80% of the fatigue life for all the tested specimens. Subsequently, it exhibits a rapid increase in the propagation stages of the fatigue crack before final failure. This behavior is different from that detected in relation to the second-order nonlinearity parameter, which was far more sensitive to fatigue damage, although, starting from 80 to 90% of the useful life in the propagation phase, both parameters are in good agreement.

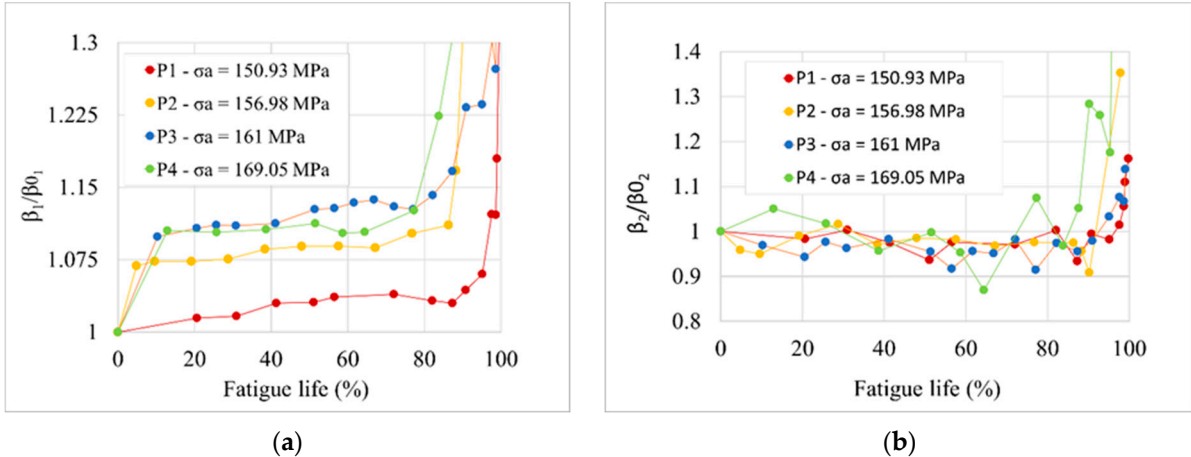

(**a**)                                                              (**b**)

**Figure 6.** Trend of the normalized second-order nonlinearity parameter $\beta_1$ (**a**) and third-order nonlinearity parameter $\beta_2$ (**b**) as a function of the fatigue life (%) of the fatigue-tested specimens.

In order to verify that the different applied stress levels were sufficient to determine the plasticization of the material near the notch for all tested specimens and to obtain a quantitative evaluation of the plastic zone, a finite numerical FEM model of the notch was built, assuming the hypothesis of the plastic behavior of the material based on a kinematic hardening law, the von Mises yield criterion and the constitutive law derived from the stress–strain curve reported by the authors in a previous paper [50]. Exploiting the geometrical symmetry, the FEM model consists of a mapped mesh of 4200 quadrilateral elements with a parabolic shape function. The element length corresponding to the notch tip was 0.1 mm and was determined on the basis of sensitivity analysis. The stress amplitude corresponding to the maximum load levels applied to the tested specimens is reported in Figure 7, where the dimension of the plastic zone is represented by the dark region near the notch tip, which was obtained by superimposing a grey color on the elements of the FEM model that were related to the plasticization phenomenon, as determined via von Mises equivalent stress. The dimension of the plastic area is also reported in Table 7, where it was observed that as the level of stress amplitude applied to the specimens increases, the corresponding percentage variation of the second-order nonlinearity parameter $\beta_1$ increases.

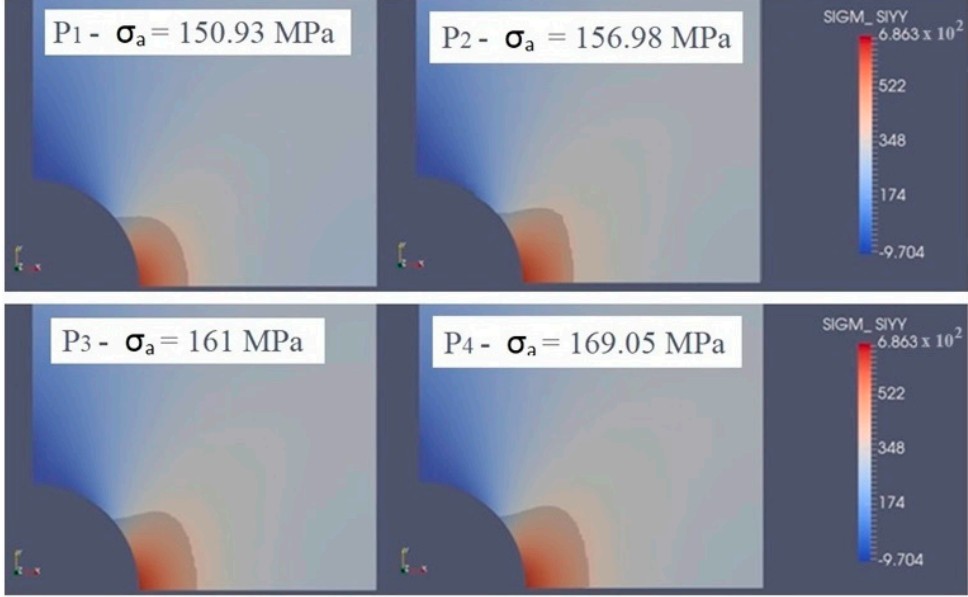

**Figure 7.** Numerical evaluation of stress state and localization of the plastic zone (dark area at the notch tip).

**Table 7.** Percentage variation of $\Delta\beta_1/\beta_{0_1}$ in the initial instant between 0 and 20% of fatigue life.

| Specimen ID | Stress Amplitude [MPa] | Percentage Variation of ($\Delta\beta_1/\beta_{01}$) [%] | Dimension of the Plastic Zone [mm²] |
|:---:|:---:|:---:|:---:|
| $P_1$ | 150.93 | 1.4 | 1.091 |
| $P_2$ | 156.98 | 6.9 | 1.339 |
| $P_3$ | 161 | 9.9 | 1.504 |
| $P_4$ | 169.05 | 10.53 | 1.934 |

Another interesting parameter to be taken into consideration when monitoring the evolution of the fatigue damage is the peak-to-peak voltage of the ultrasound signal in the time domain $\Delta V_{pp}$. Calculating this parameter at the same number of cycles of Tables 3–6, it is possible to show the attenuation of $\Delta V_{pp}$ with the number of cycles (Figure 8a), which is consistent with the behavior found by the authors in a previous work [31].

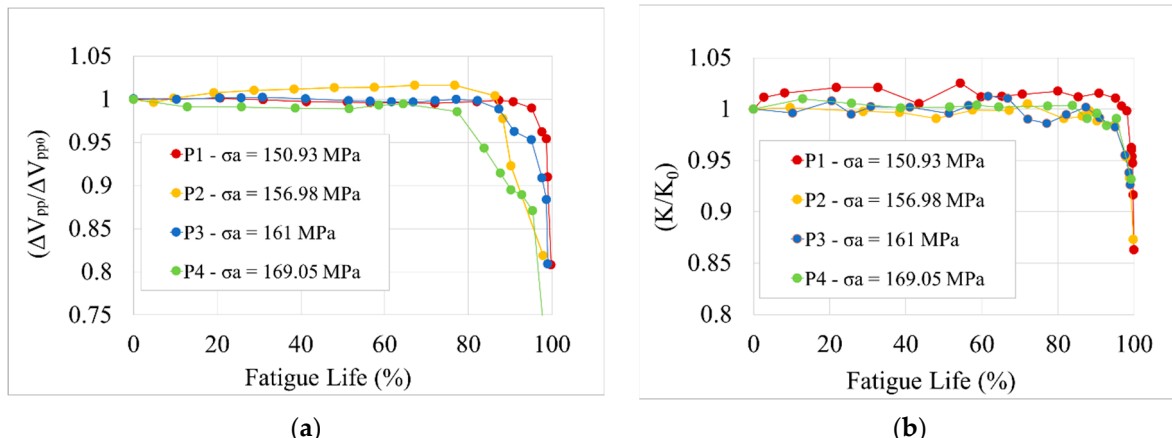

(**a**)          (**b**)

**Figure 8.** (**a**) Normalized peak-to-peak voltage $\Delta V_{pp}/\Delta V_{pp0}$; (**b**) normalized stiffness $K/K_0$ as the percentage of fatigue life varies for the tested specimens.

To make a comparison between the data collected for the tested specimens, the received signal was normalized with respect to the reference signal relating to the specimen not yet subjected to fatigue cycles ($\Delta V_{pp}/\Delta V_{pp0}$); in the same way, the number of fatigue cycles was normalized with respect to the fatigue life of the specimen (number of fatigue cycles/total fatigue life). This linear parameter generally showed a slight increase in values followed by a decrease, which became increasingly rapid in the crack propagation phase before reaching failure (Figure 8a).

The final reduction in the ultrasonic signal, justified by the presence of a crack propagation phase, is coherent with the decay of the measured stiffness during the tests. By processing the fatigue data, the stiffness normalized with respect to its initial value was plotted as a function of fatigue life (Figure 8b). The curves show a trend very similar to that of $\Delta V_{pp}$, presenting an almost constant first part and an initial slight decrease from 72% of fatigue life for the $P_3$ specimen and from 74% for the $P_1$ specimen, and then rapidly reduced to 87% of the useful life for $P_3$ specimen and 95% for $P_1$ until specimen failure. The $P_4$ specimen shows an almost stable initial stiffness value and a marked reduction from about 80% of fatigue life. On the other hand, the $P_2$ specimen exhibits a reduction in stiffness from approximately 70% of fatigue life. For all specimens tested, the normalized ultrasound signal and the normalized stiffness showed a trend that is similar to the increased fatigue life percentage. For all tested specimens, the curves reported in the graphs of Figure 9 show a fairly linear relationship between the two variables.

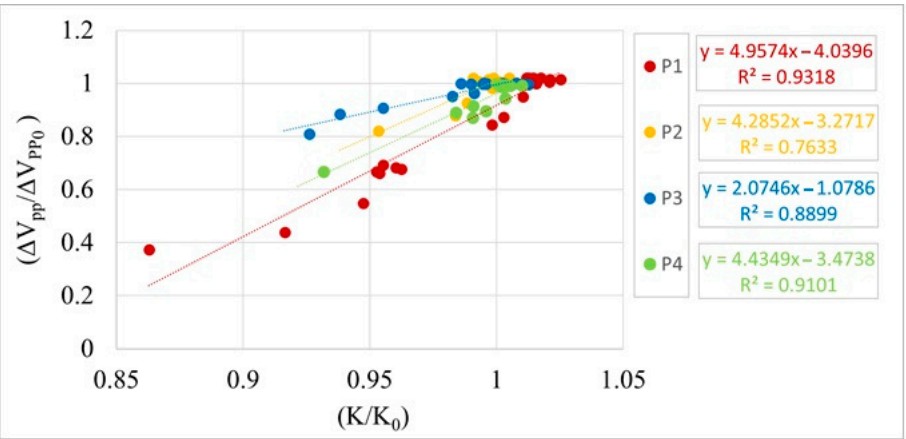

**Figure 9.** Correlation between peak-to-peak voltage of the normalized ultrasound signal and normalized stiffness for each specimen tested.

An indicator of damage *DI* was defined to obtain a quantitative assessment of the evolution fatigue damage, which is expressed by the following equation:

$$DI\ (N) = \left| \frac{\Delta V_{pp}(N) - \Delta V_{pp}(0)}{\Delta V_{pp}(0)} \right| \tag{14}$$

where

- $\Delta V_{pp}$ (N) is the peak-peak voltage of the signal recorded at *N* cycles;
- $\Delta V_{pp}$ (0) is the amplitude of the signal measured before test, referred to the undamaged specimen not yet subjected to load cycles.

The trend of the damage indicator is shown in Figure 10a for all tested specimens. The graph shows a general increase in the damage indicator starting from about 25–40% of the fatigue life for all tested specimens. In the initial stages of the test, a rapid increase in the damage index was observed for the $P_4$ and $P_2$ specimens from 0% up to about 20% of the fatigue life. Subsequently, an increase in the damage index is observed from approximately 20 to 25% of the fatigue life up to approximately 50–77% of the life. The results obtained for the $P_1$ and $P_3$ specimens, on the other hand, show a slight increase in the damage index in the initial stages of the test. Subsequently, an increase in the damage index is observed from 30 to 40% of the fatigue life up to approximately 67–72% of the useful life.

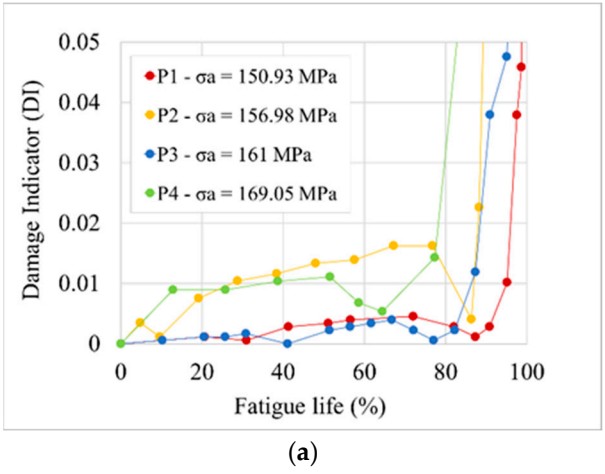

(**a**)

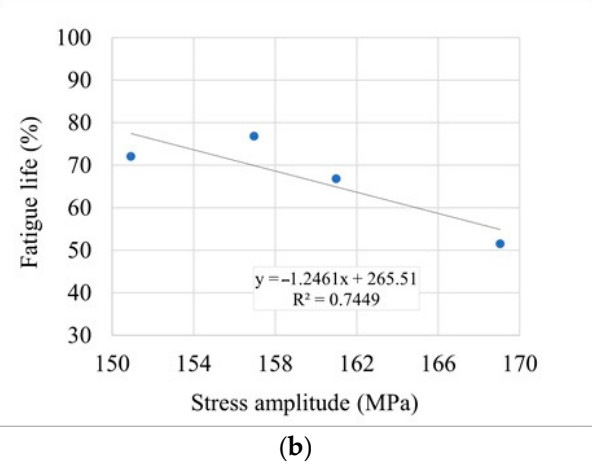

(**b**)

**Figure 10.** (**a**) Trend of damage indicator *DI* vs. fatigue life (%); (**b**) correlation between fatigue life (%) in the range 47–83%, in which DI decreases rapidly before the final rapid increase vs. stress amplitude for all tested specimens.

From the general trend of the curves, it was interesting to observe, in the opinion of the authors, a pattern repeated in the data between 47 and 83% of the fatigue life, which shows a rapid decrease in the damage indicator that occurs at lower fatigue life percentages as the applied load level increases and, therefore, also damage propagation, highlighted by the considerable increase in the damage index, occurs earlier as expected. The latter result is shown in Figure 10b, which shows the correlation between the fatigue life percentage in which the rapid decrease in the damage indicator *DI* occurs and the applied stress amplitude. The phase of final reduction in the damage indicator and the beginning of its rapid increase could coincide with the end of the nucleation phase and the beginning of crack propagation. This behavior appears to be load-dependent and repetitive, suggesting that it may represent an overall trend.

Figure 11 show two examples of fracture surface morphology with visible cracks highlighted on side A and B for the $P_3$ specimen (Figure 11a,b) and side A for the $P_4$ specimen (Figure 11c,d) observed with the stereo microscope after the fatigue failure tests.

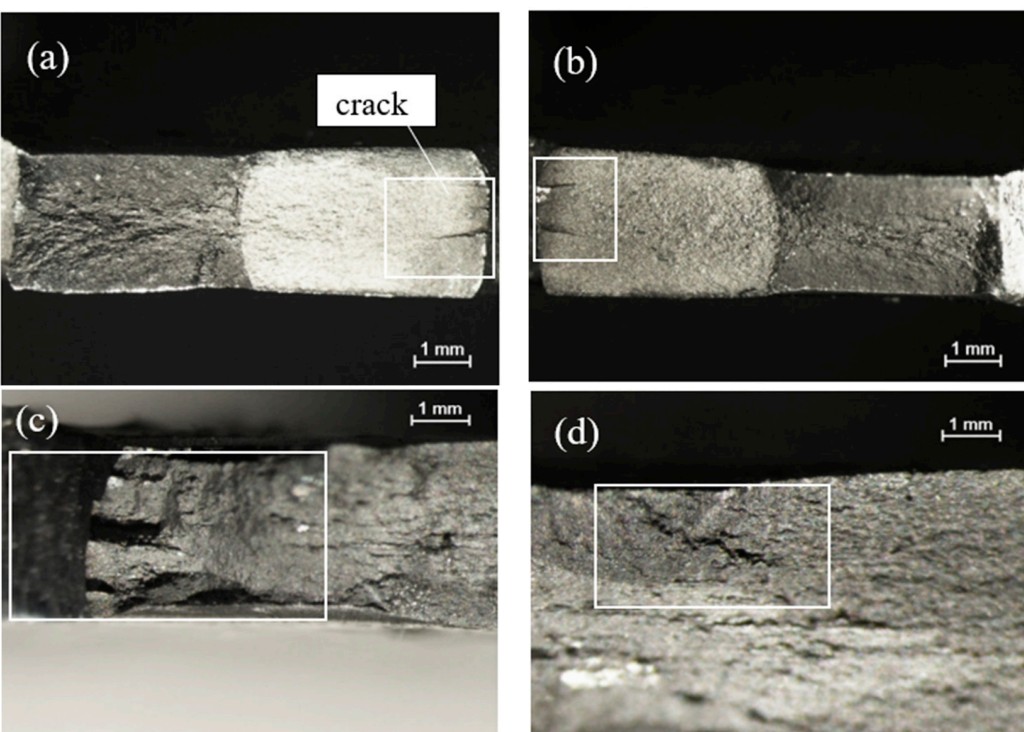

**Figure 11.** Fracture surface analysis by stereo microscope after fatigue test with visible cracks. (**a**) side A and (**b**) side B for the $P_2$ specimen; (**c**,**d**) side A for $P_4$ specimen.

## 5. Conclusions

In this paper, the change in the nonlinear acoustic properties of notched-normalized carbon steel specimens subjected to fatigue loads was proposed to evaluate the degradation and damage level of the material. The main results are summarized as follows:

(1) An increase in second-order nonlinearity was observed in the early stages of application of the load cycles from 0% to about 10–20% of the fatigue life due to the plasticization phenomena of the material at the notch tip. From approximately 30 to 40% of fatigue life, the second-order nonlinearity increases due to the progression of fatigue damage.

(2) The degradation of the material is not always accompanied by an increase in nonlinearity. In fact, a decrease in the $\beta_1$ values was observed before a rapid increase due to the accumulation of fatigue damage. A rapid increase in the $\beta_1$ parameter is observed before the final failure, starting from 85% of fatigue life, which is associated with the end of the nucleation and the beginning of the propagation.

(3) The third-order nonlinear parameter $\beta_2$ was instead less sensitive to fatigue damage, presenting a quick increase only starting from approximately 80 to 85% of the fatigue life in the crack propagation phase before final failure. The latter behavior agrees with the observed stiffness decrease.

(4) A damage indicator *DI* based on ultrasonic measurements has been proposed to follow the evolution of fatigue damage, which shows an interesting, repeated behavior between 50 and 83% of the fatigue life for all tested specimens. An approximate linear rapid decrease in this parameter has been observed, which occurs at lower fatigue life percentages as the applied load increases. Subsequently, *DI* rapidly increases in the crack propagation phase.

**Author Contributions:** Conceptualization and writing—original draft preparation, A.S. and R.N.; methodology and formal analysis, A.S.; supervision writing—review and editing, R.N. and A.S.; data curation, R.N. and A.S.; investigation, validation and visualization, A.S. All authors have read and agreed to the published version of the manuscript.

**Funding:** This research received no external funding.

**Data Availability Statement:** The data and analysis in this study are available upon request from the corresponding authors. The data are not publicly available due to limited cloud storage.

**Conflicts of Interest:** The authors declare no conflicts of interest.

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
