# Peer review of "In Situ Fatigue Damage Monitoring by Means of Nonlinear Ultrasonic Measurements"

_metals, doi:10.3390/met14010011_

Round 1

Reviewer 1 Report

Comments and Suggestions for Authors

1.  Can you show a photo of the actual specimen besides Figure 1a?

2. It is very difficult to see the peak-peak from Figure 4a. A larger scale plot should be provided. 3. Figure 5 needs to be re-formatted. The red and light blue lines are barely visible. 4. Table 3-6 is not mentioned in the main text. They should be moved to appendix. 5. Conclusions are lengthy. It should have bullet points to summarize key findings. 6. For Eq. 3, can you demonstrate how accurate this approximation will be by only keeping n=1, 2, and 3? Also there is an error in the equation, it should be beta n-1. 7. More details are needed for FEM in Fig. 7. What is the number and size of elements? What are the assumptions made in simulation? Is there a sensitivity analysis on mesh size conducted to show that results are convergent? 8. For Fig. 11, why are both sides presented for P3 but only side A for P4?

Author Response

Reviewer #1:

  1. Can you show a photo of actual specimen besides Figure 1a?

Authors’ reply:

A picture of the specimen has been added to Figure 1a.

  1. It is very difficult to see the peak-peak from Figure 4a. A larger scale plot should be provided.

Authors’ reply:

The quality of Figure 4a has been improved. Moreover, a detail of the peak-peak has been added in order to clarify the difference of the two curves.

  1. Figure 5 needs to be re-formatted. The red and light blue lines are barely visible.

Authors’ reply:

Also in this case, as suggested by reviewer, the quality of the figure has been improved and the curves are now clearly distinguished.

  1. Table 3-6 is not mentioned in main text. They should be moved to appendix.

Authors’ reply:

The text was not modified, as the introduction of Tables 3-6 was already present in the text (lines 247-250) with the following sentence: "The complete description of the experimental results is resumed in Tables 3-6. Each table is referred to a single specimen and reports the values of the three harmonics A1, A2, A3 and β1, β2 parameters for each load cycle interval." on lines 247-250.

  1. Conclusions are lengthy. It should have bullet points to summarize

Authors’ reply:

As suggested by the reviewer, the conclusion has been reformulated and summarized using a numbered list.

  1. For Eq. 3, can you demonstrate how accurate this approximation will be by only keeping n=1, 2, and 3? Also there is an error in the equation, it should be beta n-1.

Authors’ reply:

The approximation of the constitutive equation reported in Eq. 3 to the first three terms, corresponding to n = 3, is fully accurate. In fact, it is possible to evaluate the contribution of these terms dividing the Eq. 3 by the elastic stress, calculating considering only the first linear term σelastic = Eε. We obtain:

For simplicity, if we consider a reference deformation of 1000 μm/m, which is an important level of strain, and we substitute the experimental values of β1 and β2 reported in Table 3, the second and third terms become 7.3E-7 and 4.10E-12, respectively. These effects are therefore extremely limited, and it is possible to estimate that the further term would be negligible.

Finally, effectively the last coefficient β in Eq (3) was incorrect and it has been changed to βn-1. Thank you for this observation.

  1. More details are needed for FEM in Fig. 7. What is the number and size of elements? What are the assumptions made in simulation? Is there a sensitivity analysis on mesh size conducted to show that results are convergent?

Authors’ reply:

In the finite element model, the authors assumed as a hypothesis a plastic behavior of the material based on a kinematic hardening law, the Von Mises yield criterion and the constitutive law derived from the stress-strain curve reported in Figure 1b. These details were reported in the manuscript at line 300-302. However, some details cited by reviewer have been omitted in the original submission for suck of simplicity and are now inserted in the text (line 302-305):

“Exploiting the geometrical symmetry, the FEM model consists of a mapped mesh of 4200 quadrilateral elements having a parabolic shape function. The element length in correspondence of the notch tip was of 0.1 mm and has been determined on the basis of a sensitivity analysis.”

  1. For Fig. 11, why are both sides presented for P3 but only side A for P4?

Authors’ reply:

In order to give a complete description of the fracture surface, authors choose to report both sides for the first specimen, and the fracture surface at different magnification for the second specimen.

Reviewer 2 Report

Comments and Suggestions for Authors

The reviewed work entitled "In-situ fatigue damage monitoring by means of nonlinear ultra sonic measurements" contains original results of damage monitoring of C45 carbon-steel and is suitable for publication in the Metals. Despite the great efforts of the authors of the work as to the transparency of the presented results, I found some inaccuracies and errors, and some issues should be expanded. Below is a list of comments for the work:

 1. Introduction

 in the literature review, it is worth adding the influence of the initial material condition on fatigue tests. This type of research can be performed for soft and strengthened material after cold forming, characterized by strong structural defects and texture. This could be added to the literature review.

 3. Materials and Methods

 - table 1 - these are not the properties of C45 steel. C45 steel with a carbon content of approximately 0.4% has a yield strength of approximately 500 MPa and a tensile strength of approximately 800 MPa and a Yung modulus of approximately 1.7-1.8 GPa. So you either used a different type of steel, or you included the wrong table in your work. Data from table 1 indicate that it is C20 steel - 0.2%C.

 - Figure 6a shows the differences between p1-p5 and Figure 6b - in the range 0-80 it looks like these were wrong measurements. Please comment

5. Conclusion

conclusions too long. please shorten and reword them

Comments on the Quality of English Language

The level of English is good,

Author Response

Reviewer #2:

The reviewed work entitled "In-situ fatigue damage monitoring by means of nonlinear ultrasonic measurements" contains original results of damage monitoring of C45 carbon-steel and is suitable for publication in the Metals. Despite the great efforts of the authors of the work as to the transparency of the presented results, I found some inaccuracies and errors, and some issues should be expanded. Below is a list of comments for the work:

  1. Introduction

In the literature review, it is worth adding the influence of the initial material condition on fatigue tests. This type of research can be performed for soft and strengthened material after cold forming, characterized by strong structural defects and texture. This could be added to the literature review.

Authors’ reply:

The authors thank the reviewer for his/her suggestion to improve the work. In this regard, the introduction has been updated by introducing the effects of processes such as cold forming on the material. The fatigue resistance is in fact influenced by the forming process of the material, which can cause strong defects such as the distortion of the metal lattice with consequent cutting and sliding of the grains, as well as the elongation of the grains. These defects affect conventional properties such as work hardening and residual stresses of the material. Please, see the corrections in the text.

  1. Materials and Methods

 - table 1 - these are not the properties of C45 steel. C45 steel with a carbon content of approximately 0.4% has a yield strength of approximately 500 MPa and a tensile strength of approximately 800 MPa and a Yung modulus of approximately 1.7-1.8 GPa. So you either used a different type of steel, or you included the wrong table in your work. Data from table 1 indicate that it is C20 steel - 0.2%C.

Authors’ reply:

The mechanical properties indicated by referee are referred to a hardened and tempered state. In our case, the specimens were extracted by a plate in a normalized state, which material properties are coherent with the experimental static data that have been obtained and reported in Table 1. However, the indication of the state was omitted in the original submission and has now added in the text (line 168). Thank you for the observation.

 - Figure 6a shows the differences between p1-p5 and Figure 6b - in the range 0-80 it looks like these were wrong measurements. Please comment

Authors’ reply:

If authors have interpretated correctly the observation of the referee, the measurements that could appear “wrong” are referred to the data in the range 0-80 of Figure 6b. These data are calculated based on values of the β2 parameter (see Tables 3-6), which are characterized by extremely lower values. Therefore, the values are practically constant against fatigue life and produces an oscillation around zero of the ratios to the initial value. For this reason, the parameter β2 has a limited sensitivity to the fatigue damage, as already reported in the text (line 283-290).

  1. Conclusion

conclusions too long. please shorten and reword them

Authors’ reply:

As suggested by the reviewer, the conclusion has been reformulated and summarized using a numbered list.

Reviewer 3 Report

Comments and Suggestions for Authors

1)The numbers in figures and in Table 2 should be consistent.

2)Line 270: “The only difference was recorded to the initial trend of P1 specimen, which is not interested by the initial increase that characterizes the other specimens.”.Please explain the reason for this.

3)What is the purpose of Figure 11 and the authors need further explanation.

4)The conclusion of the article needs to be revised. It should be simplified or quantitatively summarized how to characterize the fatigue damage progress of metals by nonlinear acoustic measurements, rather than describing the experimental phenomena.

5)If the authors are all from the same school, there is no need to add superscripts.

Comments on the Quality of English Language

The English grammar should be double checked throughout the paper.

Author Response

Reviewer #3:

1)The numbers in figures and in Table 2 should be consistent.

Authors’ reply:

The stress amplitude was identified as σamp in Table 2 and it has now indicated as σa according to all figures. Moreover, the values of σa reported in the Table and in the figures have been checked and no discrepancy have been found.

2)Line 270: “The only difference was recorded to the initial trend of P1 specimen, which is not interested by the initial increase that characterizes the other specimens.”. Please explain the reason for this.

Authors’ reply:

Authors do not have a specific explanation for this behaviour, which characterizes the trend of the specimen evaluated at the lower stress amplitude. It could be possible that this behaviour is associated to the low level of the stress amplitude, but other experimental data are requested to confirm this hypothesis. For this reason, authors decided to highlight this experimental fact without adding any specific comment in the text.

3)What is the purpose of Figure 11 and the authors need further explanation.

Authors’ reply:

The fracture surfaces of Figure 11 have been inserted for completeness, to justify that the failure originates at notch tip and that the final failure was characterized by a significant plasticization, denoting the high ductility of the material used for the experimental work.

4)The conclusion of the article needs to be revised. It should be simplified or quantitatively summarized how to characterize the fatigue damage progress of metals by nonlinear acoustic measurements, rather than describing the experimental phenomena.

Authors’ reply:

As suggested by the reviewer, the conclusion has been reformulated and summarized using a numbered list.

5)If the authors are all from the same school, there is no need to add superscripts.

Authors’ reply:

The superscripts have been removed, according to the referee observation

Comments on the Quality of English Language

The English grammar should be double checked throughout the paper.

Authors’ reply:

The English text has been carefully revised and several changes have been introduced to improve the quality and the readability of the text.